# Association Between Diet Quality and Cardiorespiratory Fitness in Korean Adults: The 2014–2015 National Fitness Award Project

**DOI:** 10.3390/nu12113226

**Published:** 2020-10-22

**Authors:** Mingyeong Seong, Youjin Kim, Saejong Park, Hyesook Kim, Oran Kwon

**Affiliations:** 1Department of Clinical Nutrition Science, Graduate School of Clinical Health Sciences, Ewha Womans University, 52, Ewhayeodae-gil, Seodaemun-gu, Seoul 03760, Korea; 162cac05@ewhain.net; 2Department of Nutritional Science and Food Management, Ewha Womans University, 52, Ewhayeodae-gil, Seodaemun-gu, Seoul 03760, Korea; eugene841226@gmail.com; 3Nutrition Epidemiology and Data Science, Friedman School of Nutrition Science and Policy, Tufts University, Boston, MA 02111, USA; 4Department of Sport Science, Korea Institute of Sport Science, 727, Hwarang-ro, Nowon-gu, Seoul 01794, Korea; saejpark@kspo.or.kr

**Keywords:** recommended food score, diet, cardiorespiratory fitness, maximal oxygen uptake

## Abstract

Cardiorespiratory fitness (CRF) is a strong and meaningful indicator for predicting mortality, including cardiovascular disease, as well as simple physical capacity. Healthy eating is thought to be one of the crucial factors associated with an individual’s CRF status, although little research has been done on the relationship between healthy eating and CRF. This study aimed to investigate the association between overall diet quality and CRF among Korean adults. The study involved 937 adults (380 men and 557 women) aged 19‒64 years who participated in the 2014‒2015 Korea Institute of Sports Science Fitness Standards project. Diet quality was assessed by the recommended food score (RFS), and CRF was determined by maximal oxygen uptake (VO_2max_) during a treadmill exercise test. Multiple regression model analyses were stratified by age (19–34, 35–49, and 50–64 years) and sex, because both factors greatly influence CRF. After multivariate adjustment, only the 19‒34 age group in both sexes showed a positive association between RFS and VO_2max_. Additionally, when physical activity was adjusted, it was still significant in men but only marginally related in women. Our results suggest that better overall diet quality may be associated with a better CRF among young adults aged 19‒34 years in Korea.

## 1. Introduction

Cardiorespiratory fitness (CRF) is defined as the circulatory and respiratory systems’ ability to effectively inhale and exhale oxygen during physical endurance activity [1]. Several studies have reported an inverse link between CRF and health risks, including metabolic syndrome [2,3,4] and all-cause mortality [1,5,6]. Moreover, Kosola et al. reported that CRF-enhancing exercises may protect overweight men from elevated oxidized low-density lipoprotein concentration [7], a predictive biomarker of various diseases related to oxidative stress [8].

Maximal oxygen uptake (VO_2max_) is an internationally accepted parameter for measuring CRF [9]. Direct measurement of VO_2max_ is highly accurate, but it is unrealistic to collect whole population data. VO_2max_ can be measured directly by a conventional exercise test with gas analysis, and can be estimated indirectly by three modes of exercise: stepping, cycling, and treadmill running [10]. No significant difference was reported between measurement protocols [11,12], although the Bruce submaximal exercise protocol is the most widely used [13].

CRF has been negatively related to oxidative stress [1,14]. Oxidative stress is caused by an imbalance between free radical production and antioxidant defenses [15,16,17]. Despite the various mechanisms of antioxidant defense, food and nutrient intakes, such as vitamin C [18], vitamin E [19], and berry fruits [20], are well-known sources of antioxidants. However, because people intake nutrients as a meal, a combination of foods, a single food, or a single nutrient have limitations when explain the complexity of diet [21]. To determine the associations between diet and health status, overall diet quality or dietary patterns have been used [22,23,24].

The recommended food score (RFS) is a quick and simple tool to assess overall diet quality based on the frequency of consumption of foods containing large amounts of antioxidants [25]. Korean adults have shown an inverse association between the modified RFS, adapted to the Korean diet, and urinary malondialdehyde concentrations [25], one of the earliest markers of lipid oxidation [26].

Considering that CRF is associated with oxidative stress, and that the RFS contains foods linked to oxidative stress, CRF might be related to RFS. The association between diet and CRF in adolescents or diabetic patients has been investigated [27,28,29], although few studies have reported this relationship in the general adult population. Moreover, no prior publication has specifically examined the relationship between RFS and CRF. Therefore, this study aimed to determine the relationship between RFS, defined as an indicator of overall diet quality, and VO_2max_, an international parameter for measuring CRF, in Korean adults.

## 2. Materials and Methods

### 2.1. Study Design and Participants

This study applied a cross-sectional design to develop criterion-referenced health-related fitness standards for the Korea Institute of Sports Science Fitness Standards (KISS FitS) project [30,31]. The KISS FitS project was initiated in 2011 by the Korea Sports Promotion Foundation and the Ministry of Culture, Sports, and Tourism to promote health through exercise and physical and sporting activities in daily life, and is currently managed by 21 centers. It provides scientific fitness tests (for adults: body composition—body mass index (BMI); body fat percentage; waist circumference; muscular strength—grip/relative grip; muscular endurance—cross-sweetening; CRF—maximal motor load test; flexibility—sitting down and bending forward), counseling, exercise prescription, and awards certificates depending on fitness level and improvement.

This study used raw data of participants aged 19–64 years collected from 2014 to 2015 in Korea by the KISS FitS project [31]. Of 2282 participants, those aged under 19 years (*n* = 4) with a medical history (stroke, myocardial infarction, angina pectoris, osteoarthritis, osteoporosis, backache, herniated intervertebral disc, and other musculoskeletal problems) or pregnant (*n* = 190) were excluded. We also excluded participants with missing data on RFS (*n* = 974), CRF (*n* = 85), and the confounding variables (*n* = 92). The final analysis included 937 participants (380 men and 557 women). The study protocols were approved by the Institutional Review Boards at the Korea Institute of Sport Science and Ewha Womans University (KISS-201403-EFS-002-02; KISS-201504-EFS-002-01, respectively).

### 2.2. RFS

The overall diet quality was assessed using the RFS, modified by Kant et al. [32] and validated by Kim et al. [25]. Participants scored 1 point for following the three-meals-a-day pattern or consuming the recommended food or food groups at least once a week, and a score of 0, otherwise. The recommended food or food groups included in the RFS were grains (1 item: steamed rice with mixed grain), legumes (4 items: soybeans and their cooked form, miso soup/soybean paste, tofu, soymilk), vegetables (17 items: Chinese cabbage [except kimchi], spinach, lettuce, wild sesame leaf, vegetable salad/wrap with vegetables, green vegetables, bean sprouts/green bean sprouts, oyster mushrooms, mushrooms [except oyster mushrooms], pepper leaves/cooked green vegetables, crown daisy/Korean leek/water cress, cucumber, carrot/carrot juice, onion, green chili, green pumpkin, zucchini/pumpkin/zucchini juice), seaweeds (2 items: laver, kelp/seaweed), fruits (12 items: strawberry, melon, water melon, peach, banana, Japanese persimmon/dried persimmon, tangerine, pear/pear juice, apple/apple juice, orange/orange juice, grape/grape juice, tomato/tomato juice), fish (5 items: oily fish, hairtail, yellow corbina/halibut, pollack, anchovy), dairy products (3 items: low-fat milk, yogurt, cheese), nuts (1 item: peanuts/almonds/pine nuts), and tea. All individual scores were summed for the overall RFS, ranging from 0 (low diet quality) to 47 (better diet quality).

### 2.3. Assessment of CRF

CRF was assessed by a graded treadmill exercise test, according to the Bruce treadmill protocol [33] and quantified as VO_2max_ (mL/kg/min), as detailed elsewhere [30]. Briefly, the test started at a speed of 2.7 km/h and a slope of 10%. The speed (4.0, 5.5, 6.8, 8.0 km/h) and slope (12, 14, 16, 18, 20%) were increased every 3 min until exhaustion (TM55 treadmill, Quinton Cardiology Systems, Inc., Seattle, WA, USA). Heart rate was continuously monitored, and was recorded at the end of the warm-up, at each exercise stage, and after each minute of recovery (Quinton Q-Stress, Quinton Cardiology Systems, Inc., Bothell, WA, USA). The criteria for termination of the exercise were as follows: (1) rating of perceived exertion of 17 or more; (2) the heart rate did not increase even when the intensity of exercise increased; (3) 85% of the heart rate reserve was reached; (4) request for interruption by participants. The VO_2max_ (mL/min) was calculated using the Bruce equation [33]: VO_2max_ (mL/kg/min) = 6.70 − 2.82 × (1: men, 2: women) + (0.056 × exercise maintaining time (s)).

### 2.4. Biochemical Studies

Blood pressure was measured using a central aortic blood pressure device (HEM-9000AI, Omron, Japan) after sufficient rest while sitting for 15 min. The blood pressure was measured twice in total, with a 3-min interval, and the average was used. If the error between the 1st and 2nd measurements was more than 10 mmHg, a 3rd measurement was taken, and the average was used.

All participants were kept fasting for more than 12 h, and then 10 mL of venous blood was collected from the upper parietal vein using an anticoagulant-laced syringe. The collected blood was transferred to an anticoagulated and untreated tube according to each analysis item. Centrifugation was performed at 3000 rpm for 10 min, and plasma and serum were extracted, except for cellular elements, and placed in storage tubes on a dedicated rack for immediate analysis. Triglycerides (TG), total cholesterol (TC), high-density lipoprotein cholesterol (HDL-C), and low-density lipoprotein cholesterol (LDL-C) were analyzed using a dry-biochemistry analyzer (SPOTCHEM EZ SP-4430, Arkray, Inc., Kyoto, Japan). The blood sugar concentration was analyzed by an enzymatic method.

### 2.5. Covariates

The participants were interviewed by trained interviewers to obtain general information on demographic and socioeconomic characteristics, medical history, and health-related behaviors, including age, family income, marital status, smoking behavior, alcohol consumption, and RFS. The marital status was classified as married and single (including divorced/separated, widowed, and never married). Current smokers were defined as those who had smoked or quit smoking within the past 12 months. Participants who consumed alcohol more than once a month were regarded as current drinkers.

BMI was calculated as weight divided by the square of the height (kg/m^2^). The body fat percentage was estimated by the eight-polar bioelectrical impedance frequencies (Inbody 720, BioSpace, Seoul, Korea). Physical activity was assessed using a Korean version of the International Physical Activity Questionnaire (IPAQ) short-form [34]. Total metabolic equivalent task hours per week (MET-h/week) was calculated by multiplying the time spent in each activity domain by their estimated intensity based on participants’ responses to a questionnaire.

### 2.6. Statistical Analysis

All analyses were stratified by age (19–34, 35–49, and 50–64 years) and sex. Continuous variables were reported as the mean and standard deviation, and categorical variables were expressed as numbers and percentages. One-way analysis of variance, followed by Duncan’s post hoc test and a chi-square test, was used to analyze differences in general characteristics and biochemical levels between the groups. Multiple regression model analyses were conducted to examine the association between the RFS and VO_2max_. Three sequential models were evaluated: (1) Model 1 was adjusted for age and body fat percentage; (2) Model 2 was further adjusted for all variables used in Model 1, plus smoking status and drinking status; (3) Model 3 was further adjusted for physical activity. All statistical analyses were performed using SAS (ver. 9.4; SAS Institute, Cary, NC, USA). Statistical significance was set at *p* < 0.05.

## 3. Results

### 3.1. General Characteristics of the Participants

As shown in Table 1, men aged 19‒34 years had a higher BMI (*p* = 0.029), lean body mass (*p* < 0.001), and VO_2max_ (*p* < 0.001), but lower RFS (*p* < 0.001) compared with the other age groups. In women, the youngest age group had the highest lean body mass (*p* < 0.001) and VO_2max_ (*p* < 0.001), but lowest BMI (*p* < 0.001), body fat percentage (*p* < 0.001), and RFS (*p* < 0.001). Irrespective of sex, younger participants were more likely to be current smokers and consume more alcohol. Despite no significant physical activity differences among men’s age groups, younger women tended to be more sedentary than older women.

### 3.2. Biochemical Characteristics of the Participants According to Age and Sex

The youngest of both sexes had the lowest blood pressure (*p* < 0.001), diastolic blood pressure (*p* < 0.001), TG (*p* < 0.001), TC (*p* < 0.001), LDL-C (*p* < 0.001), and glucose (*p* < 0.001) levels (Table 2). Women aged 19‒34 years had significantly higher HDL-C levels than those aged 35‒49 and 50‒64 years, whereas no significant difference was found among men.

### 3.3. Association between RFS and VO_2max_ According to the Age Group

In models adjusted for age and body fat percentage (Table 3), RFS was significantly associated with VO_2max_ in the youngest of both sexes (*p* = 0.001 for men, *p* = 0.033 for women). No significant associations were found in the other age groups of both sexes. In models with additional adjustment for current smoking and drinking status, the associations between RFS and VO_2max_ remained similar (*p* = 0.002 for men, *p* = 0.034 for women). After further adjustment for physical activity, the significant association remained in men aged 19‒34 years (*p* = 0.009), while a marginally significant association was found in women of the same age range (*p* = 0.071).

## 4. Discussion

The present study determined the associations between diet quality score (measured by the RFS) and CRF (assessed by VO_2max_) in healthy Korean adults. Higher overall diet quality (RFS) was associated with CRF in relatively younger adults aged 19 to 34 years, while there was no significant association in adults aged 35 years or over.

Studies have shown the association of diet quality indices with CRF. For example, the Healthy Eating Index (HEI)-2015 is positively associated with CRF, measured by the Army Physical Fitness Test (APFT) run in healthy, active American soldiers [35]. Another study showed that a priori diet quality score was positively associated with the treadmill exercise time in European Americans aged 38‒50 years [36]. Evidence suggested that this beneficial effect may be due to the consumption of foods, such as fruits, vegetables, and whole grains, which defined diet quality. These foods are components of a Mediterranean diet, which is a dietary pattern reported to have protective effects on early osteoarthritis and hepatic steatosis, as well as muscle atrophy [37]. The RFS used in this study includes similar foods, so comparable results could be expected if other diet indices were applied.

One possible mechanism that can explain this significant association of CRF with RFS is oxidative stress. Exercise has the advantage of improving physical ability while producing oxidative stress, and oxidative stress is the leading cause of physical aging [38,39]. A previous study showed that a higher RFS represented a greater consumption of dietary antioxidants and contributed to alleviating oxidative stress [25]. The consumption of antioxidants might alleviate exercise-induced oxidative stress status and promote a favorable effect on physical health through exercise [40,41]. Therefore, it can be assumed that the significant association between RFS and CRF was related to an improved CRF through the reduction of exercise-induced oxidative stress by a high-quality diet.

In addition, the beneficial effect of the dietary fiber on inflammation may be another possible mechanism to explain the relationship between RFS and CRF. Recent studies reported inverse associations between dietary fiber intake and inflammation [22,42]. A high C-reactive protein level linked to a poor CRF, measured by estimation of VO_2max_ via an exercise tolerance test, was found in men and women aged 20‒49 years from the 1999–2002 Health and Nutrition Examination Survey [43]. However, increased consumption of dietary fiber might be beneficial for reducing the C-reactive protein level [44]. Furthermore, it has also been shown that adult men and women with a relatively better CRF, measured as VO_2max_, are more likely to adhere to dietary recommendations, such as a high fiber intake [45,46]. A Mediterranean diet, a pattern of consumption of foods high in dietary fiber, is also associated with low levels of inflammation [47]. Thus, fiber-rich foods (whole grains, fruits, and vegetables), which are important constituents of the RFS, might also be associated with better CRF.

BMI is often used as a variable to correct physical differences. BMI is an important indicator of nutritional status and is easy to measure. However, physical factors, including activity and exercise, are often not reflected by BMI. Increased physical activity contributes to a lower body fat percentage. As a result, VO_2max_ and body fat percentage have shown a stronger inverse relationship with each other than VO_2max_ and BMI [48]. For this reason, in this study, we included the body fat percentage as a control variable, instead of the BMI.

The present study showed that the association between RFS and VO_2max_ was more significant in men than women, and in the youngest group (aged less than 35 years). This might be explained by the metabolic differences. In endurance exercises, women showed increased fat oxidation and decreased oxidation of carbohydrates and proteins compared with men [49]. This metabolic difference is thought to be because men and women are physiologically different. In this regard, muscle mass and hormone differences can be considered as major factors. In general, women have less muscle mass than men, and VO_2max_ is proportional to muscle mass, so naturally, women have a lower VO_2max_ than men. In addition, women have noticeably higher levels of female sex hormones (especially estrogen) than men. Estrogen levels decrease with aging, particularly after 35 years of age, such that it is rarely secreted after menopause. Many studies have reported a relationship between estrogen levels and physical fitness, including muscle strength. A meta-analysis by Gresing et al. [50], and a study of twins [51], showed that groups treated with estrogen hormones had a greater build, muscle strength, and maximum walking speed than those who did not receive estrogen treatment. With increasing age, muscle mass decreases, which leads to a decrease in VO_2max_. A recent study found that aging occurs to a notable degree around 34 years [52]. The results of this study also showed that the biochemical status was worse in the groups aged 35–49 and 50–64 years than 19–34 years (Table 2). For the elderly, whose biochemical blood indicators are already deteriorating, it is thought that overall diet quality might not play a sufficient role in reducing oxidative stress. Yu et al. reported that the levels of physical activity in midlife and older women were important determinants of the age-related decline in VO_2max_ [53]. In other words, because muscle mass decreases with age (especially, women have lower muscle mass than men, but it may be more problematic as it decreases with aging), increasing physical activity in older adults could be a more important factor for increasing VO_2max_ than improving the diet quality. Further research is needed in the future to explore this possibility.

This study has some limitations. In particular, it could not confirm the causal relationship between RFS and CRF because of its cross-sectional design. Moreover, only the RFS was used to assess diet quality, and no instruments were used for evaluating the total intake or intake of nutrients. Nevertheless, to the best of our knowledge, this study is the first to show a favorable association between the RFS and CRF in Korean adults.

## 5. Conclusions

In conclusion, we found that a higher RFS was associated with an improved CRF among healthy Korean adults, particularly in young adults aged 19‒34 years. Research to confirm this relationship will be needed in the future. Nevertheless, our findings suggest that young adults might improve CRF by improving their diet quality. In addition, this finding will be used for the basis of further research to develop strategic plans for providing different diet guidelines according to age, sex, and physical performance.

## Figures and Tables

**Table 1 nutrients-12-03226-t001:** General characteristics of study participants according to age and sex.

Variables ^1,2^	All	19‒34 Years	35‒49 Years	50‒64 Years	*p*
Men					
*n*	380	196	92	92	
Age (years)	36.2 ± 14.5	23.9 ± 4.8 ^c^	41.8 ± 4.4 ^b^	56.7 ± 4.4 ^a^	<0.001
BMI (kg/m^2^)	25.2 ± 3.5	25.7 ± 4.0 ^a^	24.8 ± 3.1 ^b^	24.7 ± 2.4 ^b^	0.029
Body fat percentage (%)	22.7 ± 6.8	22.7 ± 8.1	22.7 ± 5.3	22.9 ± 4.9	0.957
Lean body mass (kg)	32.5 ± 4.3	34.1 ± 4.3 ^a^	31.6 ± 3.7 ^b^	30.2 ± 3.6 ^c^	<0.001
Income (10,000 won/month)					
≤200	70 (18.4)	40 (20.4)	8 (8.7)	22 (23.9)	0.049
201–400	183 (48.2)	91 (46.4)	52 (56.5)	40 (43.5)	
>400	127 (33.4)	65 (33.2)	32 (34.8)	30 (32.6)	
Marital status (*n*, %)					
Single	196 (51.6)	173 (88.3)	14 (15.2)	9 (9.8)	<0.001
Married	184 (48.4)	23 (11.7)	78 (84.8)	83 (90.2)	
Current smoker (*n*, %)	104 (27.4)	70 (35.7)	20 (21.7)	14 (15.2)	<0.001
Current drinker (*n*, %)	342 (90.0)	188 (95.9)	80 (87.0)	74 (80.4)	<0.001
Physical activity (MET-h/week)	0.3 ± 0.3	0.3 ± 0.3	0.3 ± 0.3	0.4 ± 0.3	0.210
RFS (points)	23.9 ± 9.6	22.2 ± 9.6 ^b^	24.7 ± 8.5 ^a^	26.9 ± 9.8 ^a^	<0.001
Exercise endurance time (min)	10.6 ± 1.9	11.3 ± 1.8 ^a^	10.6 ± 1.7 ^b^	9.2 ± 1.8 ^c^	<0.001
VO_2max_ (mL/kg/min)	40.0 ± 6.5	42.1 ± 6.1 ^a^	40.0 ± 5.6 ^b^	35.2 ± 5.8 ^c^	<0.001
Women					
*n*	557	102	186	269	
Age (years)	46.7 ± 12.5	26.7 ± 4.6 ^c^	41.9 ± 4.3 ^b^	57.6 ± 4.0 ^a^	<0.001
BMI (kg/m^2^)	23.5 ± 3.2	22.3 ± 3.3 ^b^	23.4 ± 3.5 ^a^	24.0 ± 2.8 ^a^	<0.001
Body fat percentage (%)	31.6 ± 6.1	29.6 ± 5.7 ^b^	30.5 ± 6.8 ^b^	33.2 ± 5.4 ^a^	<0.001
Lean body mass (kg)	21.7 ± 2.7	22.0 ± 2.9 ^a^	22.3 ± 2.9 ^a^	21.1 ± 2.3 ^b^	<0.001
Income (10,000 won/month)					
≤200	158 (28.4)	18 (17.7)	18 (9.7)	122 (45.4)	<0.001
201–400	223 (40.0)	42 (41.2)	93 (50.0)	88 (32.7)	
>400	176 (31.6)	42 (41.2)	75 (40.3)	59 (21.9)	
Marital status (*n*, %)					
Single	140 (25.1)	71 (69.6)	24 (12.9)	45 (16.7)	<0.001
Married	417 (74.9)	31 (30.4)	162 (87.1)	224 (83.3)	
Current smoker (*n*, %)	6 (1.1)	2 (2.0)	3 (1.6)	1 (0.4)	0.257
Current drinker (*n*, %)	404 (72.5)	92 (90.2)	146 (78.5)	166 (61.7)	<0.001
Physical activity (MET-h/week)	0.4 ± 0.3	0.3 ± 0.3 ^b^	0.4 ± 0.3 ^a^^,b^	0.4 ± 0.3 ^a^	0.033
RFS (points)	26.2 ± 9.0	22.9 ± 9.1 ^c^	25.3 ± 8.4 ^b^	28.0 ± 8.9 ^a^	<0.001
Exercise endurance time (min)	8.4 ± 1.6	9.5 ± 1.3 ^a^	8.6 ± 1.5 ^b^	7.8 ± 1.5 ^c^	<0.001
VO_2max_ (mL/kg/min)	29.7 ± 5.2	33.4 ± 4.3 ^a^	30.5 ± 4.8 ^b^	27.7 ± 4.9 ^c^	<0.001

^1^ BMI, body mass index; current smoker, smoking within the past 12 months; current drinker, consuming alcohol more than once a month; MET-h/week, metabolic equivalent task hours per week; RFS, recommended food score; VO_2max_, maximal oxygen uptake. ^2^ Data are presented as mean ± standard deviation or numbers (percentage). Values followed by different superscript letters are significantly different among the three groups by F test at *p* < 0.05, with Duncan’s post hoc test.

**Table 2 nutrients-12-03226-t002:** Biochemical levels of the participants according to age and sex.

Variables ^1,2^	All	19‒34 Years	35‒49 Years	50‒64 Years	*p*
Men					
*n*	380	196	92	92	
SBP (mmHg)	125.9 ± 12.2	123.5 ± 11.1 ^b^	126.4 ± 12.4 ^b^	130.8 ± 12.7 ^a^	<0.001
DBP (mmHg)	75.3 ± 12.0	69.3 ± 11.1 ^b^	80.8 ± 10.1 ^a^	82.7 ± 8.8 ^a^	<0.001
TG (mg/dL)	118.4 ± 79.0	96.7 ± 57.9 ^b^	148.8 ± 103.9 ^a^	134.3 ± 76.3 ^a^	<0.001
TC (mg/dL)	181.6 ± 39.9	167.1 ± 39.4 ^c^	203.0 ± 31.5 ^a^	191.1 ± 36.0 ^b^	<0.001
HDL-C (mg/dL)	54.1 ± 12.4	54.3 ± 12.3	55.5 ± 13.5	52.5 ± 11.4	0.264
LDL-C (mg/dL)	118.8 ± 32.5	109.5 ± 31.4 ^c^	133.3 ± 29.4 ^a^	123.9 ± 31.8 ^b^	<0.001
Glucose (mg/dL)	94.1 ± 17.3	87.3 ± 11.0 ^b^	99.4 ± 22.0 ^a^	103.2 ± 17.1 ^a^	<0.001
Women					
*n*	557	102	186	269	
SBP (mmHg)	117.5 ± 14.4	110.3 ± 10.7 ^c^	114.9 ± 13.3 ^b^	122.0 ± 14.9 ^a^	<0.001
DBP (mmHg)	73.3 ± 9.8	69.1 ± 8.9 ^c^	72.8 ± 10.0 ^b^	75.2 ± 9.5 ^a^	<0.001
TG (mg/dL)	97.6 ± 55.2	91.3 ± 54.2 ^b^	87.6 ± 58.4 ^b^	106.9 ± 51.8 ^a^	0.001
TC (mg/dL)	193.9 ± 41.8	166.2 ± 50.0 ^c^	187.9 ± 32.4 ^b^	208.4 ± 37.9 ^a^	<0.001
HDL-C (mg/dL)	65.2 ± 15.2	68.5 ± 14.5 ^a^	66.4 ± 16.0 ^a^	63.0 ± 14.6 ^b^	0.003
LDL-C (mg/dL)	121.6 ± 33.3	104.0 ± 28.5 ^c^	114.2 ± 27.6 ^b^	133.5 ± 34.1 ^a^	<0.001
Glucose (mg/dL)	95.0 ± 14.4	89.2 ± 12.5 ^c^	93.0 ± 11.4 ^b^	98.5 ± 16.0 ^a^	<0.001

^1^ SBP, systolic blood pressure (average of two measurements); DBP, diastolic blood pressure (average of two measurements); TG, triglyceride; TC, total cholesterol; HDL-C, high-density lipoprotein cholesterol; LDL-C, low-density lipoprotein cholesterol. ^2^ Data are presented as mean ± standard deviation or numbers (percentage). Values followed by different superscript letters are significantly different among the three groups by F test at *p* < 0.05, with Duncan’s post hoc test.

**Table 3 nutrients-12-03226-t003:** Multiple linear regression analysis for the association between the RFS and VO_2max_ according to age and sex.

	All				19‒34 Years				35‒49 Years				50‒64 Years			
	*n*	*β*	SE	*p*	*n*	*β*	SE	*p*	*n*	*β*	SE	*p*	*n*	*β*	SE	*p*
Men																
Model 1 ^1^	380	0.079	0.028	0.006	196	0.124	0.036	0.001	92	−0.055	0.065	0.399	92	0.099	0.060	0.103
Model 2 ^2^	380	0.067	0.029	0.020	196	0.115	0.036	0.002	92	−0.064	0.066	0.335	92	0.098	0.061	0.113
Model 3 ^3^	380	0.045	0.028	0.107	196	0.090	0.034	0.009	92	−0.064	0.068	0.348	92	0.070	0.059	0.237
Women																
Model 1 ^1^	557	0.017	0.021	0.412	102	0.081	0.038	0.033	186	−0.035	0.037	0.346	269	0.028	0.032	0.377
Model 2 ^2^	557	0.015	0.021	0.470	102	0.082	0.038	0.034	186	−0.034	0.036	0.346	269	0.023	0.032	0.480
Model 3 ^3^	557	0.008	0.021	0.704	102	0.067	0.037	0.071	186	−0.045	0.037	0.235	269	0.022	0.033	0.502

^1^ Model 1 is adjusted for age and body fat percentage. ^2^ Model 2 is adjusted for smoking and drinking status, in addition to the adjustments made in Model 1. ^3^ Model 3 is adjusted for physical activity (MET-h/week), in addition to the adjustments made in Model 2. *P-*values are from multiple regression analysis between the recommended food score (RFS) and the maximal oxygen uptake (VO_2max_). SE: standard error.

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
