# Peer review of "Association Between Diet Quality and Cardiorespiratory Fitness in Korean Adults: The 2014–2015 National Fitness Award Project"

_nutrients, 2020, doi:10.3390/nu12113226_

Round 1
Reviewer 1 Report
Physical fitness in old age is determined primarily by physical fitness achieved up to 25 years of age, physical activity in mature age, organic and functional changes resulting from past diseases and external factors affecting the human body negatively. Therefore, it is possible that an imbalanced diet may affect the VO2 max level in people of different ages. However, as could have been expected, such a relationship has not been demonstrated in the studies. However, the interpretation of these relationships based only on changes related to oxidative stress is a great simplification because the Genome-Wide Association Study; GWAS, has revealed that more than 700 genes and non-genetic areas of the genome participate in the regulation of energy metabolism, thus being responsible for a varied physiological response to exercise and weight control. Genetic factors - mainly variants of genes encoding the factors determining the energy balance of metabolic changes (including in particular enzymes involved in lipid metabolism), thermogenesis or cell differentiation processes (mainly adipogenesis), as well as genes encoding the factors modifying eating habits (mainly those controlling hunger and satiety) - play a key role in the regulation of body weight and the body's response to physical effort. Therefore, it seems that the correlation between RFS and VO2 max in the group of young men was rather accidental.
Nevertheless, the article was prepared by the authors very carefully. Research methods also do not raise any objections.
I recommend that the authors in the discussion pay more attention that by far the strongest determinant of VO2max with increasing age is the level of physical activity.
Author Response
Physical fitness in old age is determined primarily by physical fitness achieved up to 25 years of age, physical activity in mature age, organic and functional changes resulting from past diseases and external factors affecting the human body negatively. Therefore, it is possible that an imbalanced diet may affect the VO2 max level in people of different ages. However, as could have been expected, such a relationship has not been demonstrated in the studies. However, the interpretation of these relationships based only on changes related to oxidative stress is a great simplification because the Genome-Wide Association Study; GWAS, has revealed that more than 700 genes and non-genetic areas of the genome participate in the regulation of energy metabolism, thus being responsible for a varied physiological response to exercise and weight control. Genetic factors - mainly variants of genes encoding the factors determining the energy balance of metabolic changes (including in particular enzymes involved in lipid metabolism), thermogenesis or cell differentiation processes (mainly adipogenesis), as well as genes encoding the factors modifying eating habits (mainly those controlling hunger and satiety) - play a key role in the regulation of body weight and the body's response to physical effort. Therefore, it seems that the correlation between RFS and VO2 max in the group of young men was rather accidental.
Nevertheless, the article was prepared by the authors very carefully. Research methods also do not raise any objections.
I recommend that the authors in the discussion pay more attention that by far the strongest determinant of VO2max with increasing age is the level of physical activity.
-> Authors: We sincerely appreciate the reviewer’s insightful and constructive comments and suggestions. Accordingly, we supplemented the discussion section, explaining why the association between the RFS and VO2max appears only in young adults and not in older adults (Lines 233–253).
Reviewer 2 Report
The present study aimed to investigate the association between overall diet quality and cardiorespiratory fitness among Korean adults. The topic is of interest, but some corrections should been performed.
Mayor comments
It is not clear if the VO2max have been measured or if it has been estimated, please clarify in the method section and include detailed information.
In the results section, it is described the biochemical characteristics of the participants in table 2, however, this variables have not been explained in the methods section and should be included. However, this information is not used to respond to the main objective, so if it is not going to be used to justify the results it could be eliminated.
The last part of the discussion, focused on the differences obtained by sex and age, should be discussed more deeply, explaining what have been found in previous studies and the practical implications for the findings.
Minor comments
Please, include the references of the RFS in the introduction, page 2 line 53.
Method. Please, specify the treadmill equipment used during the treadmill exercise test and the equipment used to monitored the heart rate.
Results. In table 1, the superscript number 2 should be included next to 1 in the variable word. In the table, appear the superscript letter a, b, c in some data, but is has not been explained their meaning in the legend. This can also be applied to table 2. In table 2, the superscript number 1 is not correct, it is the same than table, and it has to be adapted to table 2.
In table 3, the superscript number 4 should be included in all the p.
In the discussion section, page 7, line 165, include the meaning of HEI- Healthy Eating Index
Please replace “percentage of body fat” for “body fat percentage” at page 7, line 193 and 194.
The sentence of page 7, line 195-196 should be more clear with a sentence like “For this reason, in this study, we included the percentage of body fat as a control variable, instead of the body mass index”.
Author Response
The present study aimed to investigate the association between overall diet quality and cardiorespiratory fitness among Korean adults. The topic is of interest, but some corrections should been performed.
-> Authors: We sincerely appreciate the reviewer’s insightful and constructive comments and suggestions. Please see our detailed responses below.
Mayor comments
It is not clear if the VO2max have been measured or if it has been estimated, please clarify in the method section and include detailed information.
-> Authors: Thank you for this suggestion. We described how VO2max was estimated in the methods section and added detailed information (Lines 111–112).
In the results section, it is described the biochemical characteristics of the participants in table 2, however, this variables have not been explained in the methods section and should be included. However, this information is not used to respond to the main objective, so if it is not going to be used to justify the results it could be eliminated.
-> Authors: We supplemented the results and discussion sections with a description of the participants’ biochemical characteristics according to their age groups. Therefore, without deleting Table 2, the methods for measuring these variables were added and described in the methods section (Lines 113–126 and 244–246).
The last part of the discussion, focused on the differences obtained by sex and age, should be discussed more deeply, explaining what have been found in previous studies and the practical implications for the findings.
-> Authors: Thank you for these valuable comments. Accordingly, we explained the results according to sex and age in more detail at the end of the discussion (Lines 233–253).
Minor comments
Please, include the references of the RFS in the introduction, page 2 line 53.
-> Authors: We have added a reference for the RFS in the introduction (Line 53).
Method. Please, specify the treadmill equipment used during the treadmill exercise test and the equipment used to monitored the heart rate.
-> Authors: We have specified the treadmill equipment and the equipment used to monitor the heart rate (Lines 105–108).
Results. In table 1, the superscript number 2 should be included next to 1 in the variable word.
-> Authors: This part has been modified (Table 1).
In the table, appear the superscript letter a, b, c in some data, but is has not been explained their meaning in the legend.
-> Authors: This part has been modified (Table 1).
This can also be applied to table 2. In table 2, the superscript number 1 is not correct, it is the same than table, and it has to be adapted to table 2.
-> Authors: This part has been modified (Table 2).
In table 3, the superscript number 4 should be included in all the p.
-> Authors: This part has been modified (Table 3).
In the discussion section, page 7, line 165, include the meaning of HEI- Healthy Eating Index
-> Authors: We have added the full definition of HEI (Lines 191–192).
Please replace “percentage of body fat” for “body fat percentage” at page 7, line 193 and 194.
-> Authors: This has been modified (Lines 266 and 267).
The sentence of page 7, line 195-196 should be more clear with a sentence like “For this reason, in this study, we included the percentage of body fat as a control variable, instead of the body mass index”.
-> Authors: Thank you for this suggestion. Accordingly, we have modified this sentence (Lines 228 –229).
Reviewer 3 Report
Overall considerations
Manuscript titled “Association between Diet Quality and Cardiorespiratory Fitness in Korean Adults: the 2014–2015 National Fitness Award Project” deal an interesting topic about the association between CRF and diet. Healthy eating is thought to be one of the crucial factors associated with an individual’s CRF status, this study aimed to investigate this association among Korean adults. The study 20 participants consisted of 937 adults who participated in the 2014/15 Korea Institute of Sports Science Fitness Standards project. Diet quality was assessed by the recommended food score (RFS), and CRF was determined by maximal oxygen uptake (VO2max) during a treadmill exercise test. The analysis suggest that better diet quality may be associated with a better CRF among Korean adults.
This paper is interesting and suitable with the purpose of the journal. The originality of the paper have to be highlighted even if I do not agree with Authors when it’s said that there is not anything similar already published, that is wrong since there are many researches about the association between diet and cardiorespiratory fitness about diabetes or adolescents.
Authors can comment and quote the following interesting and recent papers as follow in order to introduce a recall about Mediterranean diet:
Physical activity and Mediterranean diet based on olive tree phenolic compounds from two different geographical areas have protective effects on early osteoarthritis, muscle atrophy and hepatic steatosis. Eur J Nutr. 2019 Mar;58(2):565-581. doi: 10.1007/s00394-018-1632-2. Epub 2018 Feb 15. PMID: 29450729.
Impact of Western and Mediterranean Diets and Vitamin D on Muscle Fibers of Sedentary Rats. Nutrients. 2018 Feb 17;10(2):231. doi: 10.3390/nu10020231. PMID: 29462978; PMCID: PMC5852807.
The combined association of adherence to Mediterranean diet, muscular and cardiorespiratory fitness on low-grade inflammation in adolescents: a pooled analysis. Eur J Nutr. 2019 Oct;58(7):2649-2656. doi: 10.1007/s00394-018-1812-0. Epub 2018 Sep 3. PMID: 30178141.
Obesity, risk of diabetes and role of physical activity, exercise training and cardiorespiratory fitness. Prog Cardiovasc Dis. 2019 Jul-Aug;62(4):327-333. doi: 10.1016/j.pcad.2019.08.004. Epub 2019 Aug 20. PMID: 31442513.
The manuscript
22-29 The explanation about the methods used for the analyses should not be in the abstract section. It is enough to cite how the volunteers have been studied, avoiding the recall about p-value.
31 Consider to add “diet” among keywords.
33 In the Introduction section lacks a clear answer to the question “Which is the association between diet and cardiorespiratory fitness?” since authors are just explaining what cardiorespiratory fitness is, what VO2max is, etc. Since this is the introduction and not the material and methods section, authors should consider to explain this point in order to allow the reader to understand the aim of the study.
67-71 There is no real explanation of what “KISS FitS” is, if readers would compare the KISS program with another fitness program they cannot since there are no citation or similar to the program.
85-86 What does the number next to food groups mean? If it mean the number of permitted food, authors could report which kind food are.
87 Is the summary of individual scores showed somewhere? Otherwise readers cannot understand the RFS average score.
91 The treadmill speed, 1.7 mph, is better written in km value.
169-171 Authors are talking about the Mediterranean diet so, as mentioned before, they should consider to explain more the benefits of Mediterranean diet (publications reported in overall considerations).
182 As reported in overall considerations, authors might quote the Mediterranean effects on low-grade inflammation in adolescents.
It remains unclear what kind of physical activity volunteers did during the KISS program and what type of diet volunteers followed. Readers cannot understand the association between diet and CRF since the data about diet and physical activity are missing.
In the conclusion section please add limitations of the study and please highlight better the scientific/clinical relevance of your work. Please provide a clear message of the importance of this paper in the scientific community.
Table 1. What a/b/c apex mean? There is no subsequent link to these apex.
Table 2. Same as table 1.
Author Response
Manuscript titled “Association between Diet Quality and Cardiorespiratory Fitness in Korean Adults: the 2014–2015 National Fitness Award Project” deal an interesting topic about the association between CRF and diet. Healthy eating is thought to be one of the crucial factors associated with an individual’s CRF status, this study aimed to investigate this association among Korean adults. The study 20 participants consisted of 937 adults who participated in the 2014/15 Korea Institute of Sports Science Fitness Standards project. Diet quality was assessed by the recommended food score (RFS), and CRF was determined by maximal oxygen uptake (VO2max) during a treadmill exercise test. The analysis suggest that better diet quality may be associated with a better CRF among Korean adults.
This paper is interesting and suitable with the purpose of the journal. The originality of the paper have to be highlighted even if I do not agree with Authors when it’s said that there is not anything similar already published, that is wrong since there are many researches about the association between diet and cardiorespiratory fitness about diabetes or adolescents.
-> Authors: Thank you for these valuable comments. We agree that there have been studies on adolescents and diabetic patients. However, there are few studies on adults in the general population, so this is additionally mentioned in the introduction section (Lines 58–60).
Authors can comment and quote the following interesting and recent papers as follow in order to introduce a recall about Mediterranean diet:
-> Authors: Thank you for this valuable comment. We quoted the first and third of the papers below in the discussion section (Lines 196–198, 221).
Physical activity and Mediterranean diet based on olive tree phenolic compounds from two different geographical areas have protective effects on early osteoarthritis, muscle atrophy and hepatic steatosis. Eur J Nutr. 2019 Mar;58(2):565-581. doi: 10.1007/s00394-018-1632-2. Epub 2018 Feb 15. PMID: 29450729.
Impact of Western and Mediterranean Diets and Vitamin D on Muscle Fibers of Sedentary Rats. Nutrients. 2018 Feb 17;10(2):231. doi: 10.3390/nu10020231. PMID: 29462978; PMCID: PMC5852807.
The combined association of adherence to Mediterranean diet, muscular and cardiorespiratory fitness on low-grade inflammation in adolescents: a pooled analysis. Eur J Nutr. 2019 Oct;58(7):2649-2656. doi: 10.1007/s00394-018-1812-0. Epub 2018 Sep 3. PMID: 30178141.
Obesity, risk of diabetes and role of physical activity, exercise training and cardiorespiratory fitness. Prog Cardiovasc Dis. 2019 Jul-Aug;62(4):327-333. doi: 10.1016/j.pcad.2019.08.004. Epub 2019 Aug 20. PMID: 31442513.
The manuscript
22-29 The explanation about the methods used for the analyses should not be in the abstract section. It is enough to cite how the volunteers have been studied, avoiding the recall about p-value.
-> Authors: We deleted the p-value from the abstract (Lines 27 and 28).
31 Consider to add “diet” among keywords.
-> Authors: Thank you for this valuable comment. We added “diet” as a keyword (Line 30).
33 In the Introduction section lacks a clear answer to the question “Which is the association between diet and cardiorespiratory fitness?” since authors are just explaining what cardiorespiratory fitness is, what VO2max is, etc. Since this is the introduction and not the material and methods section, authors should consider to explain this point in order to allow the reader to understand the aim of the study.
-> Authors: Thank you for these valuable comments. The last paragraph of the introduction was revised to clarify the association between diet and CRF in the context of our study aim (Lines 45 and 57–63).
67-71 There is no real explanation of what “KISS FitS” is, if readers would compare the KISS program with another fitness program they cannot since there are no citation or similar to the program.
-> Authors: We have described the “KISS FitS” project and the variables it measured (Lines 71–74).
85-86 What does the number next to food groups mean? If it mean the number of permitted food, authors could report which kind food are.
-> Authors: We have clarified the meaning of the numbers next to the food groups and listed the food types (Lines 88–98).
87 Is the summary of individual scores showed somewhere? Otherwise readers cannot understand the RFS average score.
-> Authors: We described in more detail how to calculate individual scores, and added which foods are included in each food group. Table 1 shows the average RFS score by gender and age group (Lines 86–98, Table 1).
91 The treadmill speed, 1.7 mph, is better written in km value.
-> Authors: We modified the units of the treadmill speed to “km” (Line 104).
169-171 Authors are talking about the Mediterranean diet so, as mentioned before, they should consider to explain more the benefits of Mediterranean diet (publications reported in overall considerations).
-> Authors: Thank you for this comment. We quoted the third paper listed above in the discussion section (Lines 196–198).
182 As reported in overall considerations, authors might quote the Mediterranean effects on low-grade inflammation in adolescents.
-> Authors: Thank you for this valuable comment. We quoted the third paper listed above in the discussion section (Line 221).
It remains unclear what kind of physical activity volunteers did during the KISS program and what type of diet volunteers followed. Readers cannot understand the association between diet and CRF since the data about diet and physical activity are missing.
-> Authors: “KISS FitS” was designed to measure the physical fitness of healthy Koreans, assess their health status, and suggest appropriate levels of fitness for disease prevention. It is not intended to follow a specific physical activity or diet but is a survey of the usual physical activity, physical performance, and usual diet. The results of comparing RFS and physical activity (MET) according to age and gender are presented in Table 1. The results of examining the relationship between RFS and physical activity (MET) according to age and gender by pearson's correlation analysis are shown in Table a below. Excluding women aged 19-34 and men aged 50-64, RFS was found to have a positive association with MET.
| Table a. Correlation between the RFS and physical activity(MET) according to age and sex. | ||||||||||||
| All | 19‒34 years | 35‒49 years | 50‒64 years | |||||||||
| n | r | p | n | r | p | n | r | p | n | r | p | |
| Men | 380 | 0.175 | 0.001 | 196 | 0.147 | 0.039 | 92 | 0.245 | 0.019 | 92 | 0.156 | 0.137 |
| Women | 557 | 0.218 | <0.0001 | 102 | 0.147 | 0.141 | 186 | 0.279 | <0.001 | 269 | 0.162 | 0.008 |
In the conclusion section please add limitations of the study and please highlight better the scientific/clinical relevance of your work. Please provide a clear message of the importance of this paper in the scientific community.
-> Authors: The limitations of this study are mentioned in the last paragraph of the discussion section. As suggested in the comment, we have revised the conclusions to clarify the importance and relevance of our study to the scientific community (Lines 260–263).
Table 1. What a/b/c apex mean? There is no subsequent link to these apex.
-> Authors: We added the description of “a/b/c” (Table 1).
Table 2. Same as table 1.
-> Authors: We added the description of “a/b/c” (Table 2).
Round 2
Reviewer 2 Report
The authors have responded correctly to the required changes and the methodology of the study is now clear.
Only a minor sugestions:
Change this sentences in table 1 and 2: "Values followed by different superscript letters are significantly different among the three groups by F test at p < 0.05, with Duncan's post hoc test"
by "Values followed by different superscript letters are significantly different among those groups by F test at p < 0.05, with Duncan's post hoc test", because in some cases there are only differents between 2 groups.